# Distribution and Phylogenetic Position of the Antarctic Ribbon Worm *Heteronemertes longifissa* (Nemertea, Pilidiophora)

**Alexei V. Chernyshev** * and **Neonila E. Polyakova**

A.V. Zhirmunsky National Scientific Center of Marine Biology, Far Eastern Branch, Russian Academy of Sciences, ul. Palchevskogo 17, Vladivostok 690041, Russia
* Correspondence: nemertea1969@gmail.com

**Abstract:** To date, a total of 23 valid species of heteronemerteans belonging to 15 genera have been recorded in Antarctic and Subantarctic waters. The ribbon worm *Heteronemertes longifissa* (Hubrecht, 1887) is the only heteronemertean species reported to have bipolar distribution, but this statement is doubtful. The phylogenetic relationships of *H. longifissa* to other heteronemerteans remain uncertain. A genetic analysis of specimens from Antarctica has shown that the name *H. longifissa* refers to two sibling species with an uncorrected *p*-distance of 5.3% in COI. These species differ in body color: one is whitish, and the other is grayish-pink. The species with the whitish body has been reliably identified from off the Norway coast (as *Cerebratulus* sp. NemBar1383 (BOLD: ACM5920)), i.e., it has a bipolar distribution. A molecular phylogenetic analysis of Lineidae based on five gene markers (COI, 16S, 18S, 28S, and histone H3) has shown the genus *Heteronemertes* to belong to Lineage D of Clade 2 sensu Kajihara et al., 2022 (crown Lineidae). The phylogenetic positions of four more species of unidentified lineids are currently under discussion.

**Keywords:** benthos; nemerteans; larvae; bipolar distribution





## 1. Introduction

The phylum Nemertea currently comprises approximately 1340 species [1] inhabiting mainly the littoral and sublittoral zones of the world's oceans. Studies on Antarctic and Subantarctic nemerteans have been conducted since the late 19th century [2,3], but the most important works were published in the period from 1905 to 1985 [4–14]. Nemerteans of the class Heteronemertea have always attracted special attention as these animals reach large sizes and are most frequently found in benthic samples. Thus, *Parborlasia corrugata* (McIntosh, 1876), encountered mostly in Antarctic and Subantarctic waters, grows to 1–2 m in length. A total of 23 valid species of heteronemerteans belonging to 15 genera have been recorded from Antarctic and Subantarctic waters south of latitude 50° S [14,15]. Among them, the ribbon worm *Heteronemertes longifissa* (Hubrecht, 1887) (=*Cerebratulus longifissus*, *Lineus longifissus*) is the only heteronemertean species with bipolar distribution reported. Known from Antarctic and Subantarctic waters, this species has also been found in the North and Barents seas [16,17] and even in the coastal waters of Japan [18,19]. The specimens from Japan belong to another species [14] which has recently been described as *Corsoua takakurai* Natsumi and Kajihara, 2000 [20]. Gibson [14] assumed that the nemerteans from the Arctic seas were also misidentified as *C. longifissus*. Genetic studies could address this issue, but, of all the described Antarctic and Subantarctic heteronemerteans, sequences are currently available only for *Parborlasia corrugata* s.l. [21–23]. A substantial number of sequences were obtained from adult individuals and larvae of unidentified heteronemerteans from the coastal waters of Antarctica [22,24,25]. In the present study, we discuss the results of a molecular phylogenetic analysis of Lineidae conducted in order to clarify whether *H. longifissa* actually has a bipolar distribution, and also to identify its relationships with other heteronemerteans. The systematic positions of several unidentified heteronemertean species from Antarctica are also considered.

## 2. Materials and Methods

### 2.1. Specimen Collection

Samples were collected with a Sigsbee trawl from five localities in Antarctic waters during research cruises #79 (January and February 2020) and #87 (January–April 2022) aboard the R/V *Akademik Mstislav Keldysh* (Table 1). On board, the sediment was carefully sieved through a 1000 µm mesh screen, sorted out in seawater, and nemerteans were photographed and fixed in 96% ethanol.

**Table 1.** Specimens and sampling points of some Lineidae collected during the research cruises aboard the R/V *Akademik Mstislav Keldysh*.

| Specimen | Station | Coordinates | Depth, m | Date |
|---|---|---|---|---|
| *Heteronemertes longifissa* 11 | 6615 | 60.8879 S. 45.5342 W | 370 | 30 January 2020 |
| *Heteronemertes longifissa* 20 | 6615 | 60.8879 S. 45.5342 W | 370 | 30 January 2020 |
| *Heteronemertes longifissa* 25 | 6615 | 60.8879 S. 45.5342 W | 370 | 30 January 2020 |
| *Heteronemertes longifissa* 103 | 7371 | 61.1977 S. 47.1032 W | 1459 | 8 February 2022 |
| *Heteronemertes longifissa* 107 | 7371 | 61.1977 S. 47.1032 W | 1459 | 8 February 2022 |
| *Heteronemertes longifissa* 75 | 6615 | 60.8879 S. 45.5342 W | 370 | 30 January 2020 |
| *Heteronemertes longifissa* 78 | 6615 | 60.8879 S. 45.5342 W | 370 | 30 January 2020 |
| *Heteronemertes longifissa* 81 | 6615 | 60.8879 S. 45.5342 W | 370 | 30 January 2020 |
| *Parborlasia corrugata* 111 | 7371 | 61.1977 S. 47.1032 W | 1459 | 8 February 2022 |
| *Cerebratulus* sp. Antarctica 28 | 6614 | 60.8862 S. 45.5282 W | 367 | 29 January2020 |
| Lineidae sp. Antarctica 1 | 6615 | 60.8879 S. 45.5342 W | 370 | 30 January 2020 |
| Lineidae sp. Antarctica 5 | 6658 | 61.0387 S. 50.6875 W | 740 | 18 February 2020 |
| Lineidae sp. Antarctica 14 | 6652. | 63.2897 S. 53.6021 W | 364 | 15 February 2020 |
| Lineidae sp. Antarctica 16 | 6615 | 60.8879 S. 45.5342 W | 370 | 30 January 2020 |
| Lineidae sp. Antarctica 18 | 6615 | 60.8879 S. 45.5342 W | 370 | 30 January2020 |

### 2.2. DNA Extraction, PCR Amplification, and Sequencing

Total genomic DNA was extracted from ethanol-fixed specimens using a DNA-sorb-B nucleic acid extraction kit (AmpliSens, Moscow, Russia) and DNeasy Blood according to the manufacturer's protocol. Five markers of partial nuclear 18S rRNA (18S), 28S rRNA (28S), histone H3 (H3), and mitochondrial 16S rRNA (16S) and cytochrome *c* oxidase subunit I (COI) gene sequences were amplified from the genomic DNA. Amplification of polymerase chain reaction (PCR) was carried out using the primers listed in Table S1. PCR cycling profiles were as follows: for COI, 2 min at 94 °C, 40 cycles (40 s at 94 °C, 40 s at 50 °C and 1 min at 72 °C) and 7 min at 72 °C; for 16S, 2 min at 94 °C, 40 cycles (40 s at 94 °C, 40 s at 48 °C and 1 min at 72 °C) and 7 min at 72 °C; for 18S, 2 min at 94 °C, 40 cycles (1 min at 94 °C, 1 min at 52 °C for primer pairs Tim A/1100R and 3F/18Sbi, [1 min at 49 °C for the primer pair 18Sa2.0/9R], and 1 min at 72 °C) and 7 min at 72 °C; for 28S, 2 min at 94 °C, 40 cycles (40 s at 94 °C, 40 s at 52 °C, and 1 min at 72 °C) and 7 min at 72 °C; for H3, 2 min at 94 °C, 35 cycles (40 s at 94 °C, 40 s at 55 °C and 1 min at 72 °C) and 7 min at 72 °C.

The amplified products were purified using ExoSAP (Thermo Fisher Scientific, Waltham, MA, USA). Sequencing in forward and reverse directions was carried out on an ABI Prism 3500 Genetic Analyzers (Applied Biosystems, Waltham, MA, USA) under conditions recommended by the manufacturer, using a BigDye Terminator ver. 3.1 Cycle Sequencing Kit (Applied Biosystems) and the same primers as for PCR. BLAST searches [26], as implemented in the NCBI website (http://www.ncbi.nlm.nih.gov accessed on 17 February 2023), were conducted to check for putative contamination.

### 2.3. Phylogenetic Analysis

The sequences for the five gene markers (16S, 18S, 28S, COI, and H3) were aligned separately using MAFFT ver. 7 [27] with default parameters. We put more weight on unedited alignment including all positions, as suggested by [28]. A supermatrix with a total length of 6656 bp was formed by concatenating the five markers using SequenceMatrix [29], wherein external gaps were coded as 'missing data'. Simultaneous selection of partition

schemes and the search for optimal nucleotide substitution models for the supermatrix obtained were carried out using PartitionFinder [30,31] with implementation of the 'greedy' search scheme. According to the best-suggested scheme, the final supermatrix was divided into seven partitions (Table S2). A combined analysis based on the five concatenated gene markers was conducted using Bayesian inference (BI) and maximum likelihood analyses (ML). BI was carried out in MrBayes 3.2 [32] by launching two parallel runs with four Markov chains in each run (three cold and one hot) during 10,000,000 generations.

The values of run convergence indicated that a sufficient number of trees and parameters were sampled. Based on the convergence of likelihood scores, 25% of sampled trees were discarded as burn-in. The rest was used to build the consensus tree, while the nodes with posterior probabilities of less than 50% collapsed. The maximum likelihood phylogenetic tree was inferred using the edge-147 linked partition model on the IQ-TREE web server [33]; branch supports with the 1000 ultrafast bootstrap replicates were obtained in the IQ-TREE software [34]. The BI topology was chosen as the main phylogenetic scheme for the present study. A total of 89 taxa of Lineidae from different genera were included in the molecular phylogenetic analyses, with *Sonnenemertes cantelli* and *Baseodiscus mexicanus* used as outgroups (Table 2).

We reconstructed the haplotype network based on the COI gene sequences including both all original *H. longifissa* sequences and one accessed from GenBank trimmed to the length of the shortest sequence, 600 bp, using TCS v1.21 software under 95% connection limit. The haplotype network was visualized as a pie chart with geographic information taken into account using the tcsBU web-based program.

The uncorrected pairwise *p*-distances were calculated in MEGA ver. 6.0.

**Table 2.** List of species included in the phylogenetic analysis of Lineidae s.l., with GenBank accession numbers for sequences (the sequences new to this study are highlighted in bold).

| Species | 16S | 18S | 28S | COI | H3 | Source |
|---|---|---|---|---|---|---|
| *Apatronemertes albimaculosa* | JF277587 | JF293030 | HQ856860 | HQ848584 | JF277733 | [21] |
| *Cerebratulus lacteus* | JF277575 | JF293044 | HQ856857 | HQ848576 | JF277728 | [21] |
| *Cerebratulus marginatus* | JF277576 | JF293042 | HQ856858 | HQ848575 | JF277729 | [21] |
| *Cerebratulus orochi* | LC538101 | LC538103 | LC538104 | LC538102 | LC538105 | [35] |
| *Cerebratulus mordukhovichi* | OM422971 | OM423090 | OM423029 | OM456681 | OM468125 | [36] |
| *Cerebratulus* sp. NemBar1383 | – | – | – | KP697728 | – | Strand unpubl. |
| ***Cerebratulus* sp. Antarctica28** | **OQ449306** | **OQ449292** | **OQ449324** | **OQ450482** | **OQ446609** | **Present study** |
| *Cerebratulus* sp. DH-2009 isolate A4pilidia04 | GU227009 | | | GU227125 | | [24] |
| Cf. Heteronemertea sp. DH-2009_isolate_D4pilidia04 | GU227013 | – | – | GU227120 | – | [24] |
| Cf. Heteronemertea sp. DH-2009_isolate_A4larva04 | GU227014 | – | – | GU227126 | – | [24] |
| *Corsoua takakurai* | LC520112 | – | LC520126 | LC520106 | LC520128 | [20] |
| *Dushia wijnhoffae* | EF124878 | – | EF178494 | EF124967 | – | [37] |
| *Dushia* cf. *nigra* | LC389832 | LC389840 | LC389844 | LC389867 | LC389851 | [37] |
| *Euborlasia maycoli* | LC520114 | LC520121 | LC520125 | LC520108 | – | [38] |
| *Gorgonorhynchus albocinctus* | – | LC010650 | LC010651 | LC010649 | – | [39] |
| *Gorgonorhynchus* cf. *bermudensis* | KF935467 | KF935300 | KF935356 | KF935517 | KF935412 | [40] |
| *Gorgonorhynchus* cf. *repens* | LC520115 | LC520122 | LC520123 | LC520105 | LC520131 | [41] |
| Heteronemertea gen. sp. 4 TCH-2015_isolate_119 | KU197548 | – | KU365690 | KU197835 | – | [42] |
| Heteronemertea sp. 17 | LC625672 | LC625688 | LC625699 | LC625640 | LC625729 | [43] |
| ***Heteronemertes longifissa* 11** | **–** | **OQ449293** | **OQ449325** | **OQ450483** | **–** | **Present study** |
| ***Heteronemertes longifissa* 20** | **OQ449307** | **OQ449294** | **–** | **OQ450484** | **OQ446610** | **Present study** |
| ***Heteronemertes longifissa* 25** | **OQ449308** | **OQ449295** | **OQ449326** | **OQ450485** | **OQ446611** | **Present study** |
| ***Heteronemertes longifissa* 75** | **–** | **–** | **–** | **OQ450486** | **–** | **Present study** |
| ***Heteronemertes longifissa* 78** | **OQ449309** | **OQ449296** | **OQ449327** | **OQ450487** | **OQ446612** | **Present study** |
| ***Heteronemertes longifissa* 81** | **–** | **–** | **–** | **OQ450488** | **–** | **Present study** |
| ***Heteronemertes longifissa* 103** | **–** | **–** | **–** | **OQ450489** | **–** | **Present study** |
| ***Heteronemertes longifissa* 107** | **–** | **–** | **–** | **OQ450490** | **–** | **Present study** |
| ***Hinumanemertes kikuchii* *** | **OQ449310** | **OQ449297** | **OQ449328** | **OQ450491** | **OQ446613** | **Present study** |
| *Kulikovia alborostrata* | KU821503 | - | KU856679 | KU821529 | KU821552 | [44] |
| *Kulikovia manchenkoi* | KU821497 | KY468934 | KU856671 | KU821523 | KU821546 | [44] |

**Table 2.** *Cont.*

| Species | 16S | 18S | 28S | COI | H3 | Source |
|---|---|---|---|---|---|---|
| *Kulikovia* cf. *montgomeryi* | OM422978 | OM423098 | OM423037 | OM456685 | OM456685 | [36] |
| *Kulikovia torguata* LtUr1 | KU821486 | KY468935 | KU856673 | KU821511 | KU821534 | [44] |
| Lineidae KuramBio1 12-5 | MN211473 | MN211375 | MN211427 | MN205497 | MN205448 | [45] |
| Lineidae KuramBio2 85 | MN211475 | MN211377 | MN211429 | MN205498 | – | [45] |
| Lineidae KuramBio2 77 | MN211481 | MN211383 | MN211434 | MN205504 | MN205454 | [45] |
| Lineidae sp. 41DS | MF512050 | MF512076 | MF512102 | – | MF512144 | [45] |
| Lineidae sp. Antarctica 1 | OM422984 | OM423104 | OM423043 | OM456687 | OM468138 | [36] |
| Lineidae_sp. Antarctica 5 | OM422988 | OM423108 | – | OM456691 | OM468142 | [36] |
| Lineidae sp. Antarctica 14 | OM422986 | OM423106 | OM423045 | OM456689 | OM468140 | [36] |
| Lineidae sp. Antarctica 16 | OM422987 | OM423107 | OM423046 | OM456690 | OM468141 | [36] |
| Lineidae sp. Antarctica 18 | OM422985 | OM423105 | OM423044 | OM456688 | OM468139 | [36] |
| Lineidae sp. G06 Bering | OM422979 | OM423099 | OM423038 | – | OM468133 | [36] |
| Lineidae sp. H02 IceAGE | OM422982 | OM423102 | OM423041 | – | OM468136 | [36] |
| Lineidae sp. KGK-2 | LC625651 | – | – | LC625624 | LC625709 | [43] |
| Lineidae sp. KGK-4 | LC625653 | LC625683 | – | LC625626 | LC625711 | [43] |
| Lineidae sp. KGK-6 | LC625656 | – | LC625690 | LC625627 | LC625714 | [43] |
| Lineidae sp. KGK-7 | LC625657 | – | LC625691 | – | – | [43] |
| Lineidae sp. KGK-8 | LC625658 | – | – | LC625628 | LC625715 | [43] |
| *Lineidae* sp. Kuril O11 | OM422989 | OM423109 | OM423047 | OM456692 | OM468143 | [36] |
| *Lineus acutifrons* | JF277573 | JF304778 | HQ856855 | GU590937 | JF277727 | [21] |
| *Lineus clandestinus* | **MK064103** | **OQ449298** | **OQ449329** | **MK078739** | **OQ446614** | [46] **present study** |
| *Lineus flavescens* | KP682165 | – | EF178497 | KP682050 | – | [47,48] |
| *Lineus lacteus* | JF277584 | JF293065 | HQ856850 | HQ848583 | JF277725 | [21] |
| *Lineus longissimus* | MW073006 | KY468932 | MW077245 | KY561813 | KY606234 | [44,49] |
| *Lineus sanquineus* | KF935468 | KF935301 | KF935301 | KF935518 | KF935413 | [40] |
| *Lineus ruber* | MK064093 | KY468933 * | KY468929 * | MK078684 | KY606235 * | [44,46] |
| *Lineus* sp. Guam | KU821507 | – | KY468928 | – | KY561818 | [44] |
| *Lineus viridis* | MK064101 | **OQ449299** | **OQ449330** | MK078733 | **OQ446615** | [46] **present study** |
| ***Maculaura aquilonia* **** | **OQ449311** | **OQ449300** | **OQ449331** | **OQ450492** | **OQ446616** | **Present study** |
| ***Maculaura* sp. *** | **–** | **OQ449301** | **OQ449332** | **OQ450493** | **OQ446617** | **Present study** |
| *Micrura bathyalis* | MN211479 | MN211381 | MN211432 | MN205502 | – | [45] |
| ***Micrura bella* ***** | **OQ449312** | **OQ449302** | **OQ449333** | **OQ450494** | **OQ446618** | **Present study** |
| *Micrura callima* | MN211472 | MN211374 | MN211426 | MN205496 | MN205447 | [45] |
| *Micrura chlorapardalis* | KF935459 | KF935292 | KF935348 | KF935512 | KF935404 | [40] |
| *Micrura dellechiajei* | KF935461 | KF935294 | KF935350 | KF935514 | KF935406 | [40] |
| *Micrura fasciolata* | JF277585 | JF293038 | HQ856846 | HQ848578 | JF277721 | [21] |

**Table 2.** *Cont.*

| Species | 16S | 18S | 28S | COI | H3 | Source |
|---|---|---|---|---|---|---|
| *Micrura ignea* **** | **OQ449313** | **OQ449303** | **OQ449334** | **OQ450495** | **OQ446619** | **Present study** |
| *Micrura purpurea* | JF277577 | JF293036 | HQ856845 | HQ848586 | JF277726 | [21] |
| *Micrura rubramaculosa* | KF935460 | KF935293 | KF935349 | KF935513 | KF935405 | [40] |
| *Micrura* sp. albocephala | KU197574 | – | KU365712 | KU197849 | – | [42] |
| *Micrura* sp. dark | KU197586 | – | KU365713 | KU197858 | – | [42] |
| *Micrura* sp. 3 | KU197563 | – | KU365710 | KU197841 | – | [42] |
| *Micrura* sp. 4 | KU197581 | – | KU365711 | KU197857 | – | [42] |
| *Micrura* sp. IceAGE | OM422994 | OM423114 | OM423052 | OM456696 | OM468146 | [36] |
| *Micrura* sp. IZ 132532 | KF935457 | KF935290 | KF935346 | KF935510 | KF935402 | [42] |
| *Micrura* sp. IZ 132529 | KF935458 | KF935291 | KF935347 | KF935511 | KF935403 | [40] |
| *Micrura verrilli* | KF935455 | KF935288 | KF935344 | KF935508 | KF935400 | [40] |
| *Micrura wilsoni* | KU197535 | – | KU365716 | KU197827 | – | [42] |
| *Notospermus geniculatus* | KF935462 | KF935295 | KF935351 | – | KF935407 | [40] |
| *Notospermus mitellatus* | LC625660 | LC625685 | LC625693 | LC625629 | LC625717 | [43] |
| *Nipponomicrura* sp.* | **OQ449314** | **OQ449304** | **OQ449335** | **OQ450496** | **OQ446620** | **Present study** |
| *Nipponomicrura uchidai* | KU821509 | – | KY468930 | KY561815 | KY561819 | [44] |
| *Parborlasia corrugata* | JF277578 | JF293037 | HQ856851 | EU194826 * | JF277732 | [21,22] |
| *Parborlasia corrugata* **111** | **OQ449315** | **OQ449305** | **OQ449336** | **OQ450497** | **OQ446621** | **Present study** |
| *Parvicirrus dubius* | AJ436830 | – | AJ436885 | AJ436940 | – | [50] |
| *Polydendrorhynchus zhanjiangensis* | MT659662 | MT648831 | MT648832 | MT648511 | MT655749 | [41] |
| *Pseudomicrura afzelii* | GU445914 | GU445924 | GU445919 | GU392013 | – | [51] |
| *Riseriellus occultus* | JF277581 | JF293031 | HQ856848 | HQ848581 | JF277724 | [21] |
| *Siphonenteron nakanoi* | LC625678 | – | LC625706 | LC625646 | LC625737 | [43] |
| *Tenuilineus bicolor* | AJ436823 | – | EF124960 | AJ436933 | AJ436980 | [48,50] |
| *Zygeupolia rubens* | JF277574 | JF293045 | HQ856861 | HQ848585 | JF277735 | [21] |
| *Yininemertes pratensis* | KY274025 | KY274047 | KY274069 | KY274003 | KY274091 | [40] |
| Outgroups | | | | | | |
| *Baseodiscus mexicanus* | KF935449 | KF935281 | KF935337 | KF935503 | KF935393 | [40] |
| *Sonnenemertes cantelli* | MF512048 | MF512073 | MF512099 | MF512118 | MF512141 | [45] |

* Sea of Japan, Vostok Bay; ** Sea of Okhotsk, Magadan; *** Sea of Japan, Spokoynaya Bay; **** Panama.

## 3. Results

### 3.1. Phylogenetic Analysis of Lineidae

The combined aligned sequences comprise 636 bp for 16S rDNA, 1935 bp for 18S rDNA, 3096 bp for 28S rDNA, 658 bp for COI, and 331 bp for histone H3. They contain 2449 variable sites, of which 1640 are informative; the frequencies of variable and informative sites are 36.8% and 24.6%, respectively. The frequencies of variable sites in the aligned sequences of mitochondrial 16S rDNA (63.2%) and COI (50.9%) are greater than those in sequences of nuclear 18S rDNA (26.8%), 28S rDNA (34.4%) and histone 3 (38.4%).

The general topology of the lineid phylogenetic tree is as follows: *Micrura ingnae* + (Lineage A + Lineage C + (Clade 2 + Lineage B)) (Figure 1). Clade "2 + B" is strongly supported (PP = 1, BS = 99%). Relationships of Lineage D with other Lineages of Clade 2 sensu Kajihara et al., 2022 remain unresolved; Clade 3 has low support. Three strongly supported clades, each including two or more Lineages, can be distinguished within Clade 2 (crown Lineidae). Clade N + P (PP = 1, BS = 90%) comprises representatives of Lineages N and P, and also *Hinumanemertes kikuchii*. The monophyly of Lineage N sensu Kajihara et al., 2022 is not confirmed, since *Lineus acutifrons* and *H. kikuchii* form a strongly supported subclade, whose relationships inside Clade N+P remain unclear. Clade 4 (PP = 1, BS = 93%) comprises Lineages E, H, G, I, J, K, and Q, and also Lineidae KuramBio2 77. Clade J + K (PP = 1, BS = 93%) comprises representatives of sister Lineages J and K. *Micrira dellechiajei* forms a strongly supported subclade (PP = 1, BS = 100%) with *Notospermus* species (Lineage E). Some of the lineids are not included in any of the previously identified Lineages: Lineidae sp. H02 IceAGE, Lineidae KuramBio2 77, Heteronemertea sp. 17 HA-2021, Heteronemertea gen. sp. 4 TCH-2015, *Pseudomicrura afzelii*, and *Zygeupolia rubens*.

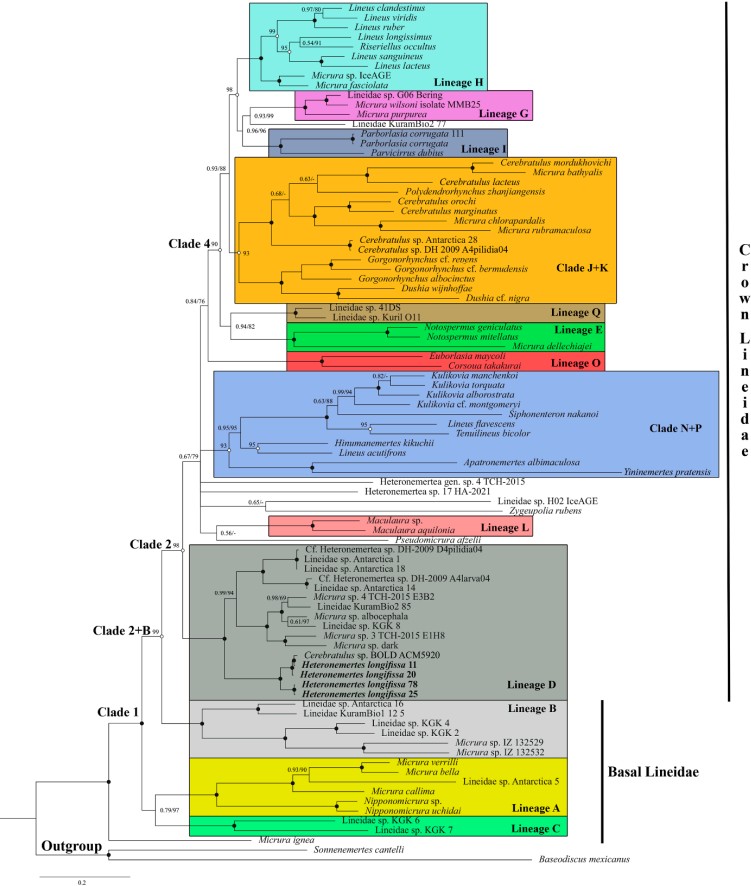

**Figure 1.** Bayesian inference (BI) phylogenetic tree for the supermatrix Lineidae of the five-marker dataset (16S, COI, 18S, 28S, H3). Numerals near the branches are nodal support values (Bayesian posterior probability/ML bootstrap value). Black circles indicate nodal support of 0.1/100; white circles indicate nodal support of 1.0/. Antarctic *Heteronemertes longifissa* specimens are highlighted in bold.

### 3.2. Antarctic Lineids

All eight individuals identified on the basis of external characters as *Heteronemertes longifissa* belong to Lineage D and are sisters of the rest of the representatives of this clade. Their distinctive feature is the very long cephalic horizontal slits. Three individuals (*H. longifissa* 25, 78, and 81) have a grayish-pink body color (Figure 2a–d), while five individuals (*H. longifissa* 11, 20, 75, 103, and 107) have a whitish color; reddish brain and beige gut are visible through translucent body wall (Figure 2e–g). Haplotypes for the COI gene are grouped into two networks (A and B) (Figure 3). The uncorrected *p*-distances between networks A and B are 5.2–5.7% in COI; the *p*-distances between the samples within each of the networks amount to 0.2–0.7% in COI (Table S3). The uncorrected *p*-distances between *H. longifissa* A and B are 3.0% and 1.3% in 16S and histone H3, respectively.

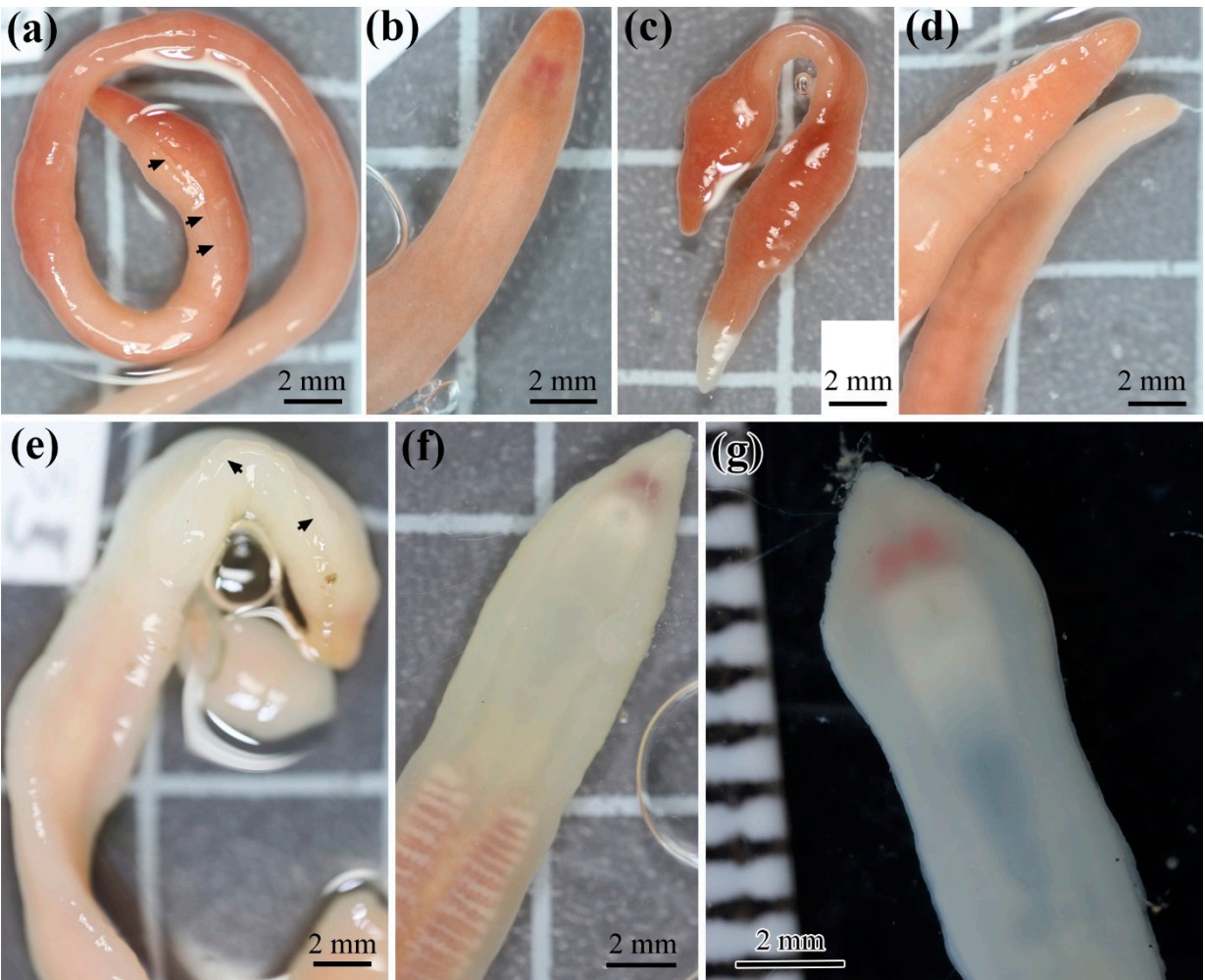

**Figure 2.** *Heteronemertes longifissa*: specimens 25 (**a**,**b**), 78 (**c**), 81 (**d**), 11 (**e**), 20 (**f**), and 107 (**g**). Arrows indicate cephalic horizontal slits.

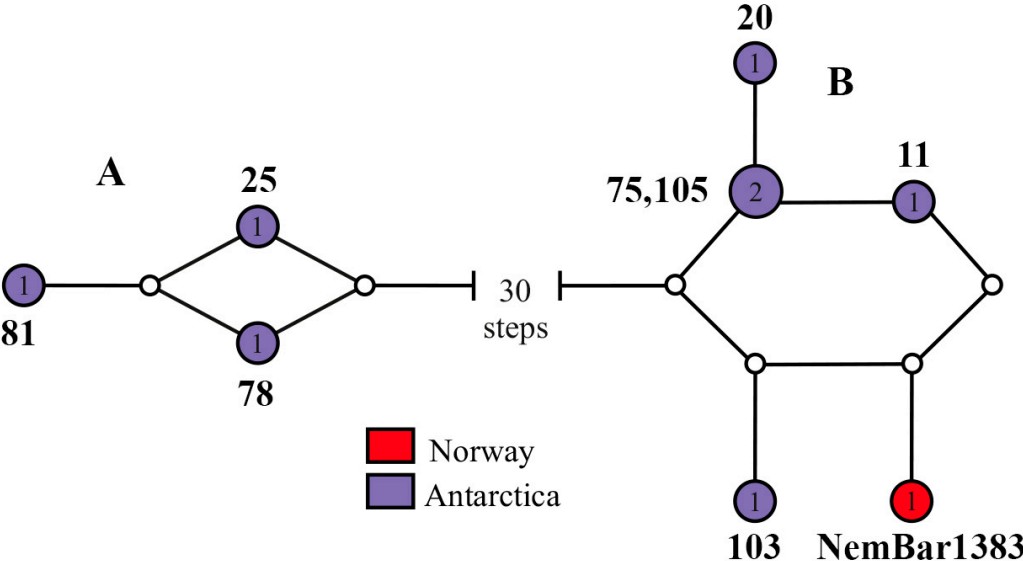

**Figure 3.** Statistical parsimony haplotype network based on the mitochondrial DNA cytochrome c oxidase subunit I gene of the examined *Heteronemertes longifissa* specimens, colored in accordance with their geographic distribution. The connecting limit is set at 95%. The line connecting the haplotype pie charts represents a single mutational change; each white dot on the line represents one additional mutational change. The numerals within the pie charts represent the number of specimens within each haplotype. The numerals near haplotypes correspond to the specimens' numbers (Table 2).

In GenBank (NCBI) and BOLD (The Barcode of Life Data System), three sequences are available that match the sequences of *H. longifissa*: two specimens of unidentified heteronemertean Nemertea sp. Antarctic ARM-2008 (accession nos. EU718394.1 and EU718388.1) from Antarctica [22] (with the uncorrected *p*-distance between this sample and *H. longifissa* B being 0.2% in 16S) and a specimen *Cerebratulus* sp. NemBar1383 BOLD: ACM5920 (accession no. KP697728) from Norway, belonging to network B (Figure 3) (with the uncorrected *p*-distances between this sample and *H. longifissa* B being 0.5–0.8% in COI—see Table S3).

The three unidentified heteronemerteans (Lineidae spp. Antarctica 1, 14, and 18) belong to Lineage D (Figure 1). Lineidae sp. Antarctica 1 and 18 have a pale olive body color with a reddish brain visible through a translucent body wall (Figure 4a,b); the body of Lineidae spp. Antarctica 14 is pale beige with a reddish brain (Figure 4c,d). A BLAST analysis has shown that the sequences of COI of pilidia from the Ross Sea under the names DH-2009 GU227120 isolate D4pilidia04 and DH-2009 GU227126 isolate A4larva04 [24] are identical to the sequences of Lineidae sp. Antarctica 1 (=Lineidae sp. Antarctica 18) and Lineidae sp. 14, respectively (Figure 1). Lineidae sp. Antarctica 16 with the deep, rose body (Figure 4e,f) is closely related to Lineidae KuramBio1 12 5 from the abyssal plain adjacent to the Kuril–Kamchatka Trench (Lineage B) (Figure 1). Lineidae sp. Antarctica 5 with a yellowish body and orange-yellowish head (Figure 4g,h) belongs to Lineage A and the subclade of the "*Micrura*" (*Evelineus*) species with a fragile soft body and red or orange anterior head end (Figure 1). *Cerebratulus* sp. Antarctica 28 with a grayish-pink body (Figure 4i–k) belongs to Lineage J (Figure 2). The pilidium from the Ross Sea under the name *Cerebratulus* sp. DH-2009 isolate A4pilidia04 [24] is conspecific to *Cerebratulus* sp. Antarctica 28 (Figure 1). One specimen with a grayish-red body and a light-colored transverse band on the head (Figure 4l) is conspecific to *Parborlasia corrugata*.

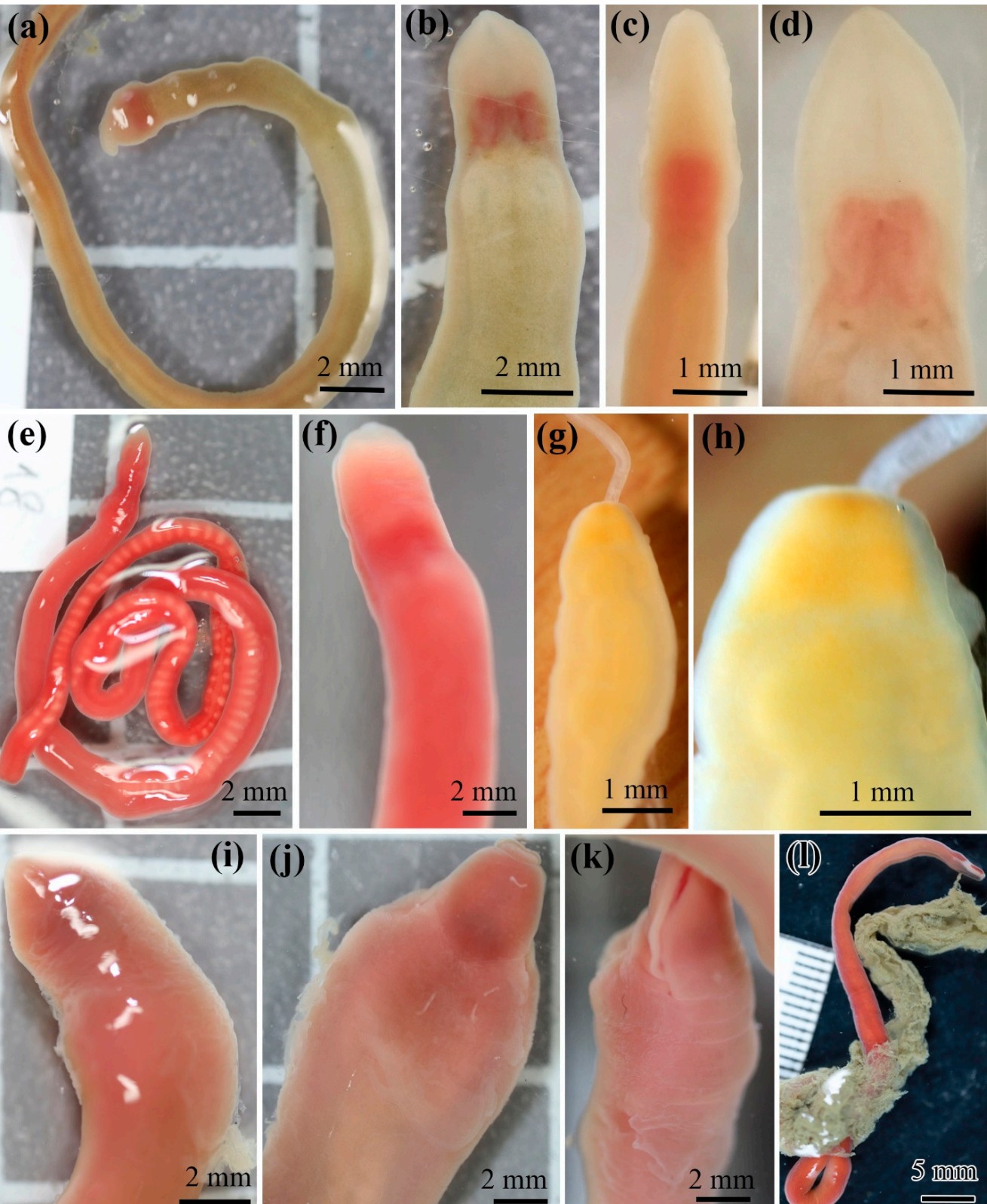

**Figure 4.** Antarctic lineids: Lineidae sp. Antarctica 1 (**a**,**b**), Lineidae sp. Antarctica 14 (**c**,**d**), Lineidae sp. Antarctica 16 (**e**,**f**), Lineidae sp. Antarctica 5 (**g**,**h**), *Cerebratulus* sp. Antarctica 28 (**i**–**k**), and *Parborlasia corrugata* 111 (**l**).

## 4. Discussion

### 4.1. Phylogenetic Analysis of Lineids

The molecular phylogenetic analyses of heteronemerteans carried out in 2019–2022 revealed the major clades within this order [36,43,45,52]. Chernyshev and Polyakova [36] proposed the names *Lineus*, *Cerebratulus*, and *Siphonenteron* for the three clades without providing definitions for them. An extensive phylogenetic analysis by Kajihara et al. [43] made it possible to identify 16 phylogenetic lineages (high-supported clades) designated with letters (A, B, C, D, etc.) within the family Lineidae. In addition, three large clades were identified: Clades 1, 2, and 3 [43]. Another Lineage (Q) was identified later [36]. The results obtained by Kajihara et al. [43] are largely confirmed by our studies with some changes and supplements. Inside Clade 2 (we suggest referring to it as "crown Lineidae"), our analysis does not support Clade 3 which should be a sister of Lineage D. In the analysis by Kajihara et al. [43], *Lineus acutifrons* belongs to Lineage N, holding a basal position in it. In our analysis, *L. acutifrons* and *Hinumanemertes kikuchii* form an independent subclade within Clade N + P. We have established the phylogenetic position of *H. kikuchii* for the first time, and this brackish-water species has not got in Lineage P that includes the freshwater species *Apatronemertes albimaculosa* and the brackish-water *Yininemertes pratensis*. We assume that these differences result from the fact that Kajihara et al. [43] included many species for which 1–2 sequences are known in their phylogenetic analysis, while for our analysis we selected a smaller number of species with 3–5 sequences known.

### 4.2. Systematic Position of the Genus Heteronemertes

Identification of heteronemerteans based on fixed specimens is challenging since in formalin and alcohol they usually lose the external traits characteristic of the species (body color pattern, head shape, etc.). In this regard, *Heteronemertes longifissa* has a characteristic feature that is clearly visible in both live and fixed individuals: very long lateral head slits reaching far behind the mouth [9,14] (Figure 2a,e). Our specimens have been identified on the basis of this feature. We have described the body color of live individuals for the first time.

The systematic position of *Heteronemertes longifissa* remains unclear. This species was described as a member of the genus *Cerebratulus* [53] and then transferred to the genus *Lineus* [9]. Gibson [14] described in detail the internal structure of *H. longifissa* and confirmed its affiliation to the genus *Lineus*, although the proboscis organization in this species is not typical of the species of *Lineus* s. str. Chernyshev [54] transferred *H. longifissa* into a new genus, *Heteronemertes*, and added such traits as the presence of very long head slits to its diagnosis. Our phylogenetic analysis has shown that *H. longifissa* belongs to Lineage D and is not closely related to either *Lineus* s. str. (Lineage H) or *Cerebratulus* s. str. (Lineage J). It is currently the only described species for which affiliation to Lineage D has been proven. Below is the revised and supplemented diagnosis of the genus *Heteronemertes*.

Genus *Heteronemertes* Chernyshev, 1995.

Type species: *Heteronemertes longifissa* (Hubrecht, 1887).

Diagnosis. Body flattened, caudal cirrus absent, and horizontal cephalic slits reach far behind the mouth. The cutis glands are not separated from the body wall musculature by a connective tissue layer. Longitudinal muscle plate between rhynchocoel and foregut absent. Proboscis typical heterotype (see [55]) with outer longitudinal musculature well developed, muscle crosses present, and neural sheath not developed into separate nerves. Foregut with subepithelial glands. Cerebral ganglia have outer and inner neurilemma; neurochords and neurochord cells are absent.

### 4.3. Is Heteronemertes longifissa a Bipolar Species?

*Heteronemertes longifissa* was first described from materials collected off Marion Island [39] and then found in different parts of the Antarctic and Subantarctic regions [9,10,13,14]. However, this species under the name *Cerebratulus longifissus* was reported for the North and Barents seas [16,17], and also as *Lineus longifissus* for the coastal waters of Japan [18,19].

The reason why the Japanese specimens were assigned to *Lineus longifissus* remains unclear, since the horizontal cephalic slits do not reach behind the mouth in them. It has been found that *Lineus longifissus* sensu Takakura, 1898 belongs to the recently described species *Corsoua takakurai*, which is attributed to Lineage O [20].

Friedrich [56] suggested that *Cerebratulus longifissus* from off Norway and the Barents Sea " … may be *Lineus longifissus*, or perhaps *Cerebratulus fissuralis*" (p. 18). *Cerebratulus fissuralis* Friedrich, 1958 from off Iceland, was described extremely briefly and without illustrations; this species differs from *Heteronemertes longifissa* by the presence of a neurochord. *Cerebratulus longifissus* sensu Punnett, 1903 from off Norway differs from *Heteronemertes longifissa* by having a thick layer of connective tissue between the dermal glands and the body wall outer longitudinal muscles, a short rhynchocoel, and a distinct cephalic vascular loop [16]. Therefore, Gibson [14] assumed *Cerebratulus longifissus* sensu Punnett, 1903 and *C. longifissus* sensu Uschakov, 1928 to belong to an independent species.

The molecular genetics analysis has shown significant genetic differences between the eight individuals of *H. longifissa* that we studied: uncorrected *p*-distances between the two *H. longifissa* networks of 5.2–5.7% in COI vs. 0.2–0.7% between the samples in each of the networks. These individuals also differ in body color: *H. longifissa* 11, 20, 75, 103, and 107 have a whitish color vs. a grayish-pinkish color in *H. longifissa* 25, 78, and 81. The low values of the COI uncorrected *p*-distances between networks A and B are a boundary between interspecific and intraspecific distances ("barcoding gap") recorded for heteronemerteans (4–9%) [57]. All three individuals were collected at the same station (Table 1) and, therefore, there is every reason to assume that *H. longifissa* includes two sibling species that differ in body color. Because of the lack of data on the body color pattern of the type specimens, the question as to which of the specimens should be assigned the name *H. longifissa* still remains open.

According to genetic data, *Cerebratulus* sp. NemBar1383 (BOLD: ACM5920), collected off Norway at a depth of 40 m north of Tromsø, is conspecific with *H. longifissa* 11, 20, 75, 103, and 107 from Antarctica. All these individuals have a whitish body color with a reddish brain visible through a translucent body wall. The snow-white body color was described also for *Cerebratulus longifissus* from off Norway [16]. The color pattern of the specimen from the Barents Sea is unknown [17]. Thus, one of the two sibling species of *H. longifissa* s.l. has a bipolar distribution. This is the first case where the bipolar distribution of nemerteans has been proven genetically. In addition to *H. longifissa*, bipolar distribution was reported for the hoplonemertean *Nipponnemertes pulchra* (Johnston, 1837) [58], but this conclusion needs genetic confirmation. A recent phylogenetic analysis has shown that *Nipponnemertes* sp. 17 from Antarctica, which looks very similar to *N. pulchra*, is an independent species [36].

### 4.4. Problems of Identification of Antarctic Heteronemerteans

One of the major challenges in the study of Antarctic and Subantarctic heteronemerteans is their species identification. First, this is explained by the fact that a substantial portion of the species was described only from fixed specimens. Thus, Gibson [14] described seven new heteronemertean species from Antarctica and Subantarctica, but none of the descriptions had information about the color and head shape of live individuals. Recent studies using gene markers show the importance of these traits in discriminating sibling species of heteronemerteans [44,46,47,52,59]. Using only internal morphology traits does not guarantee accurate identification of the sibling species. Even such a unique for heteronemerteans trait as very long horizontal cephalic slits is present in two sibling species, which is evidenced by our studies. Another issue is the great number of cryptic species among nemerteans. As has been shown recently, *Parborlasia corrugata*, widely distributed in Antarctica and the Subantarctic region, includes two cryptic species that differ genetically [22,23]. Thus, it is unclear which of these two cryptic species should be assigned the name *Parborlasia corrugata*. The same applies to *Heteronemertes longifissa*.

The problems in the identification of Antarctic heteronemerteans have resulted in a situation where sequences were obtained for only one described species (*Parborlasia*

*corrugata*) [21–23] and a large number of species unidentified to date [22,25,36], including the four Lineidae species from our study. Most of these unidentified species are probably new to science, but their description is unlikely to be possible in the following decade. In addition to the sequences of adult heteronemerteans, sequences of heteronemertean larvae have also been obtained. However, researchers note great differences in the species diversity of adult and larval samples [25]. Even a common Antarctic species such as *Parborlasia corrugata* was found to comprise only 4.3% of larvae sampled [25]. A similar phenomenon of significant differences in the species composition of larvae and adult heteronemerteans has been reported for the Pacific coast of North America [42]. In this regard, the identity of the sequences of three heteronemerteans' larvae from Antarctica with the sequences of Lineidae sp. Antarctica 1, Lineidae sp. Antarctica 18, Lineidae sp. 14, and *Cerebratulus* sp. Antarctica 28 is of certain interest. These three adult lineids were collected from a depth of 370 m. The significant differences between pilidia and adult heteronemerteans can apparently be explained by the fact that a large number of larvae of the deep-sea species, which have been studied much more poorly than shallow-water nemerteans, are found in plankton.

**Supplementary Materials:** The following supporting information can be downloaded at https://www.mdpi.com/article/10.3390/w15040809/s1, Table S1: List of primers used in the present study. Forward primer sequences are highlighted in bold; Table S2: The main parameters of model-based inferred phylogenies; Table S3: Uncorrected COI *p*-distances (%) within *Heteronemertes longifissa*.

**Author Contributions:** Conceptualization, A.V.C.; methodology, A.V.C. and N.E.P.; software, N.E.P.; validation, A.V.C. and N.E.P.; formal analysis, A.V.C. and N.E.P.; investigation, A.V.C. and N.E.P.; resources, A.V.C. and N.E.P.; data curation, A.V.C.; writing—original draft preparation, A.V.C. and N.E.P.; writing—review and editing, A.V.C.; visualization, A.V.C. and N.E.P.; supervision, A.V.C.; project administration, A.V.C.; funding acquisition, A.V.C. All authors have read and agreed to the published version of the manuscript.

**Funding:** This research was funded by the Russian Science Foundation (grant no. 22-24-00184).

**Data Availability Statement:** The datasets studied during the present study are available from the corresponding author upon reasonable request.

**Acknowledgments:** The authors wish to thank Anna E. Vlasenko and Grigorii V. Malykin for providing the material from Antarctica. We are also grateful to Evgeniy P. Shvetsov for proofreading the English of this manuscript.

**Conflicts of Interest:** The authors declare no conflict of interest.

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
