# Peer review of "Distribution and Phylogenetic Position of the Antarctic Ribbon Worm Heteronemertes longifissa (Nemertea, Pilidiophora)"

_water, doi:10.3390/w15040809_

Round 1

Reviewer 1 Report

The manuscript "Distribution and phylogenetic position of the Antarctic ribbon worm Heteronemertes longifissa (Nemertea, Pilidiophora)" discuss the phylogenetic positioning and relationship of the heteronemertean Heteronemertes longifissa. It presents evidence of two cryptic species, based on molecular and external traits, as commonly found in nemerteans. This is a very nice and well written manuscript of an elusive group within Nemertea, which is by itself, understudied as well. Also brings clarity on the difficulties on the relationships within lineids. I have only two comments:

1. I believe that the lineage discussion within the group should be approached more carefully, as no species delimitation or any statistical approach was performed here (as it would not be appropriated due to the small number of representatives per species/lineage). While there still a lot of work to do with this group, this is certainly a study that provides some insights on the needed revision on Heteronemertes and Micrura genera, in particular.

2. The haplotype network (Fig 3) is really difficult to understand. The legend is incomplete, the numbers in the circles are the sample number? If so, it would be better to create a side legend indicating the sample size represented. It would be better if we could relate the samples in each network with the individuals from the tree. Maybe using patterns or different colors to make the identification possible.

Author Response

Reviewer #1

  1. I believe that the lineage discussion within the group should be approached more carefully, as no species delimitation or any statistical approach was performed here (as it would not be appropriated due to the small number of representatives per species/lineage). While there still a lot of work to do with this group, this is certainly a study that provides some insights on the needed revision on Heteronemertes and Micrura genera, in particular.

Response: We deleted phylogenetic “part” in the Abstract and reduced phylogenetic discussion.

  1. The haplotype network (Fig 3) is really difficult to understand. The legend is incomplete, the numbers in the circles are the sample number? If so, it would be better to create a side legend indicating the sample size represented. It would be better if we could relate the samples in each network with the individuals from the tree. Maybe using patterns or different colors to make the identification possible.

Response: We modified both Figure 3 and its legend.

Reviewer 2 Report

Dear authors:

Your manuscript is very good, for me it needs only very minor changes, specifically complement the introduction with recent references.

I have these question about methodology:

- ¿why do you fix the specimens with 10% 56 formalin and 96% ethanol?, the formalin can affect the DNA for molecular analysis, normally for molecular analysis it used 96 % ethanol. If your arguments are based on literature I suggest cite it.

I have this suggestion for methodology (and future papers): 

- For study softbodies specimens, it is suggested anesthetize the specimens (in example with carbonated water or sodium sulfate) as previous condition for fix with formol and/or ethanol. For study Annellids it is very good methodology because it relax many soft structures such as parapods and prosboscis.

Many success and blessings !!!

Author Response

Reviewer #2

  1. ¿why do you fix the specimens with 10% 56 formalin and 96% ethanol?, the formalin can affect the DNA for molecular analysis, normally for molecular analysis it used 96 % ethanol. If your arguments are based on literature I suggest cite it.

Response: We deleted 10% formalin (this phrase was mistakenly got from our previous article).

2.      For study softbodies specimens, it is suggested anesthetize the specimens (in example with carbonated water or sodium sulfate) as previous condition for fix with formol and/or ethanol. For study Annellids it is very good methodology because it relax many soft structures such as parapods and prosboscis.  

Response: Unfortunately, both collectors didn’t use anesthetize. This is acceptable since we have not studied the internal morphology of these nemerteans.